# Personal and Psychological Traits of University-Going Women That Affect Opportunities and Entrepreneurial Intentions

**DOI:** 10.3390/bs14010066

**Published:** 2024-01-18

**Authors:** Luis Enrique Valdez-Juárez, Elva Alicia Ramos-Escobar, José Alonso Ruiz-Zamora, Edith Patricia Borboa-Álvarez

**Affiliations:** Department of Business and Economics Sciences, Technological Institute of Sonora Mexico, Guaymas 85000, Mexico; elba.ramos@itson.edu.mx (E.A.R.-E.); jose.ruiz@itson.edu.mx (J.A.R.-Z.); edith.borboa@itson.edu.mx (E.P.B.-Á.)

**Keywords:** personal traits, psychological profile, female university entrepreneurship

## Abstract

The purpose of this research is to analyze how personality traits and psychological profiles influence the detection of entrepreneurial opportunities by, and the intentions of, university-going women in the northwest region of Mexico. It also examines how business opportunities are decisive when it comes to awakening entrepreneurial intention. The moderating and mediating effects of the detection of business opportunities and the psychological profile are also examined with respect to the direct relationship between personal traits and entrepreneurial intentions. For this study, information was collected from 1197 students attending the Autonomous University of Baja California and the Technological Institute of Sonora through a digital survey (Google Forms) distributed via email during the second half of 2022. The PLS-SEM statistical technique was used to test the hypotheses of the proposed theoretical model. The results revealed that personality traits have positive and significant effects on the psychological profile and on business opportunities. However, it was clearly observed that one’s personal traits and psychological profile have little or no influence on entrepreneurial intentions. We also found that the psychological profile is the construct that most influences business opportunities. In addition, it was also highlighted that business opportunities contribute to awakening the entrepreneurial intentions of university-going women. On the other hand, it was revealed that business opportunities have a negative moderating effect on the relationship between the psychological profile and entrepreneurial intentions. Likewise, this study has shown that the detection of business opportunities and the psychological profile have indirect effects on the relationship between the personal traits and the entrepreneurial intentions of university-going women. This research contributes to the development and strengthening of trait theory, the theory of reasoned action, and the theory of planned behavior.

## 1. Introduction

Entrepreneurship and business activities are considered the backbone of any economy due to their great contributions to the gross domestic product and the generation of jobs, thereby achieving greater stability in economic indicators, which leads to strengthening the growth of the economy [1,2,3]. However, today, the roles of women in the academic and business fields have experienced significant evolution in recent decades, challenging deep-rooted stereotypes and reflecting a profound social transformation of the perception of gender roles [4]. The growing presence of university-going women entrepreneurs is a phenomenon that challenges traditional notions about women’s professional and educational activities, contributing a unique dimension to the intersection between education and entrepreneurship [5]. This phenomenon reflects a paradigmatic change in gender dynamics, evidencing the capacity of women to lead business initiatives while pursuing their academic studies [6].

The convergence between the university environment and entrepreneurship has opened new possibilities for women, allowing them to not only acquire academic knowledge but also to develop entrepreneurial skills from an early age [7]. This trend reflects a dynamic response to changing labor market demands and challenges conventional perceptions regarding the balance between higher education and entrepreneurship [8]. Female entrepreneurship has gained recognition as a key driver of economic growth and innovation [9]. Despite progress, women continue to face unique challenges in the business world, such as understanding the factors that contribute to the success of university-going women entrepreneurs [10]. 

The entrepreneurial university-going woman not only aspires to achieve academic goals but also seeks to make a significant contribution to the business world [11]. The intersection between higher education and female entrepreneurship creates a unique context through which to analyze personal and psychological traits as individual characteristics that shape the experiences and decisions of female entrepreneurs and drive them to embark on the challenging journey of entrepreneurship while immersed in higher education [12].

To understand the complexity of female university student entrepreneurship, a literary analysis of the subject and the related personality traits is relevant [13]. In analyzing pioneering studies on women entrepreneurs, the importance of psychological factors within the entrepreneurial process is highlighted [14]. Traditionally, for these studies, the model of the five elements that make up personality has been used. This framework helps to understand how the behavioral traits—openness to experience, awareness, extroversion, agreeableness, and emotional stability—can influence the decision making and perseverance of entrepreneurial university-going women [15], and this model has been the driving force behind several empirical and theoretical studies [16,17]. Examining these factors that manifest themselves in female entrepreneurs during their university studies allows us to obtain a greater understanding of their entrepreneurial psychology, which can manifest either positively or negatively, and how it strongly influences the success or failure of the entrepreneurial intention. For example, self-efficacy plays a crucial role in entrepreneurial behavior [18]. The belief in one’s own ability to initiate and carry out entrepreneurial actions can be decisive in the success of an entrepreneur [19], thus making it a highly relevant factor in how it influences the decision making and persistence of entrepreneurial university-going women in the face of business and academic challenges [20]. 

Taking Personality Trait Theory as a reference, certain persistent individual traits can influence a person’s behavior and decisions in various situations in the context of female university entrepreneurship. Therefore, personality traits represent a unique dimension of individuals that predict a particular type of behavior in different contexts and/or situations over a precise time [21,22]. In the context of the entrepreneurial attitude, the theory of entrepreneurial personality reveals that entrepreneurs who start and manage a new business require the fulfillment of certain very specific roles and activities in addition to particular traits, such as innovation, risk taking, personal relationships, and goal setting [23,24]. In addition, this theory provides a valuable perspective through which to determine certain personal traits that can influence the inclination and success of women in starting and managing businesses while pursuing university studies, such as self-efficacy, awareness, openness to experience, extroversion, emotional stability, persistence, leadership, and adaptability [25]. On the other hand, the dark personality in entrepreneurship is present in the personality traits, and these can interrupt these intentions. According to Koehn et al. [26] and Hmieleski and Lerner [27], these dark traits are also known as the Dark Triad, which is composed of the three following malevolent, ego-centered personality traits: narcissism, psychopathy, and Machiavellianism. These all represent socially aversive patterns of behavior that tend to manifest and be associated with highly negative outcomes. Similarly, it is important to note that social, economic, environmental, and health shocks, such as the COVID-19 pandemic, have strongly affected the emotional health, attitudes, and behaviors of women entrepreneurs [28]. However, these effects are more profound in emerging economies and in countries with developing economies [29,30]. Traditionally, the theory of planned behavior (TBP) is one of the most widely used by researchers to analyze entrepreneurial intentions. However, there is great variety and divergence in the literature regarding this behavioral phenomenon. For example, Shapero and Sokol [31] developed what they called the “Business Event” (BE) model, which, conceptually, is very similar to the content of the TPB. This BE model equated intention with the identification of a credible and personally viable opportunity. In short, for a perceived individual opportunity to be credible, the decision maker has to perceive it as desirable (the attitudes and social norms established in the TPB) and feasible (essentially, self-efficacy). In addition, another antecedent, the propensity to act, was incorporated, which captured the potential for a credible opportunity to become an intention and, subsequently, the execution of an action. In short, the TPB explains or defines that when individuals face a variety of problems related to alternative opportunities, they can act in two ways (negatively or positively), with a prior evaluation of behavior as a background. Therefore, intentions are determined by internal factors (motivation, will, reasoning, and self-efficacy, which are all personal traits) that influence rational and planned behavior [32].

Based on the theory of reasoned action (TRA), behavioral intentions can be determined based on the attitude towards the behavior, as well as subjective norms [33]. Intentions are the best predictors of planned behavior [34]. The theory of planned behavior is usually an extension of the theory of reasoned action, including perceived behavioral control (PCC), which is considered an additional antecedent of behavior and intentions [35]. The study of entrepreneurship had its foundation mainly in the TPB, which plays a fundamental role because it sheds light on the factors that influence the decisions and performances of women when undertaking and managing a business during their university studies [36,37]. Factors that are linked to the theory include the following: 1. attitude towards entrepreneurship, 2. subjective norms, 3. perceived behavioral control, 4. past experiences, 5. institutional support, and 6. entrepreneurial education [38,39]. 

In this context, the Global Entrepreneurship Monitor [40] in its report Women Entrepreneurship: Challenging Bias and Stereotypes establishes that almost 1 in 3 entrepreneurs at the heads of organizations are women (0.80 women for every 1 man). Globally, women are more likely than men to be individual entrepreneurs (1.47 female entrepreneurs for every 1 male). Globally, one in six women express the intention to start a business in the near future. The highest rates of entrepreneurial intention are observed in low-income countries (approximately 28% of women express the intention to start a business). Every tenth female entrepreneur, worldwide, is in the early stages of starting a business. Women’s creation rates are particularly high in low-income countries and in Latin America and the Caribbean, and women represent one in four high-growth entrepreneurs globally, with higher proportions in developing and low-income countries, as well as North America. 

Particularly in Mexico, as in countries with unbalanced and weak economies, economic and social indicators have had significant impacts in recent times. According to data issued by the OECD [41], it was estimated that Mexico’s economy will grow by 2.5% next year and by another 2% in 2025. For decades in this country, there have been serious economic and social problems that have plunged it into poverty, a result that is associated with inequality, a lack of opportunities, informal employment, and insufficient investment in education at all levels. Current data issued by the INEGI [42] note that of the total population in Mexico available to work, 55.0% of men and 56% of women are informally employed; furthermore, 3.4% of men and 2.8% of women are unemployed. Given this background, higher-education institutions are trying to provide a boost by shifting towards entrepreneurial education to strengthen business capabilities in youth; however, this work is insufficient and with little coverage, given that these efforts have been concentrated more on private entities than public institutions. The economic, social, and cultural gaps, particularly in relation to gender, remain substantial, not only in the field of education but also in the workplace. PwC [43] data indicate that in Mexico, women’s income represents only half of men’s estimated income in 2022, which means that there are still important labor, business, and economic barriers to gender parity. Women, globally, continue to fight for economic, social, and legal independence. However, during the COVID-19 pandemic, the problems of excelling in the economy, employment, and entrepreneurship became further aggravated by mass unemployment, childcare needs, and domestic work [44]. Data issued by UNESCO [45] indicate that Mexico reported the lowest female labor force participation rate as well as the widest gap between male and female participation rates. From Mexican universities, business incubators and entrepreneurship centers have been promoted as isolated initiatives to contribute to the development of business skills. These strategies are disjointed from public policies, but they must be linked given that they are key components for business ecosystems to detonate an inclusive economy and reduce the gender gap [46]. The increase in youth demographics, the increase in unemployment in many countries, the changes in the employment market and in the economy, and the appearance of new technologies are just some of the reasons why it is necessary to prepare future generations with entrepreneurial skills and mentalities that will allow them to respond to uncertain, demanding, and dynamic changes [45]. Ultimately, both in Mexico and in other countries, inclusive entrepreneurial education is the way to promote self-esteem and confidence based on individual capabilities and attitudes by imparting relevant skills and values that help students broaden their perspectives and opportunities [44]. Entrepreneurial education aims to strengthen (1) personal development, thus strengthening resilience and motivation; (2) economic development, due to the creation of self-employment, the reorientation of business culture towards entrepreneurship, and the introduction of disruptive innovations in the economy; and (3) social development, through the implementation of innovative ventures that help improve the quality of life of communities and address future uncertainties for life [45,47].

According to the literature review, the following approaches have been explored: 1. Do personality traits influence the psychological profiles of female university students? 2. Are personality traits and psychological profiles contributing attitudes that influence behavior towards the perception of opportunity detection and the entrepreneurial intentions of female university students? 3. Do the detection of opportunities and the psychological profile have moderating and mediating effects on the entrepreneurial intentions of female university students? In turn, this study has generated the following objectives: 1. Analyze the effects that personality traits have on the psychological profiles of female university students for the detection of entrepreneurial opportunities. 2. Analyze the influence that personality traits and psychological profiles have on the detection of business opportunities and the entrepreneurial intentions of female university students. 3. Examine the influence of the detection of business opportunities on the intentions that female university students have for the development of entrepreneurial endeavors. 4. Analyze the moderating effects of business opportunities on the relationship between personal traits and psychological profiles with the entrepreneurial intentions of female university students. 5. Examine the indirect effects (multiple mediation analysis) that entrepreneurial opportunities and psychological profiles exert on the direct relationship between the personal traits and the entrepreneurial intentions of female university students. This research has notable relevance in the following two aspects: 1. The topics addressed in the literature under analysis have garnered significant interest given that, with this research model, strategic actions can be derived to strengthen the business opportunities and entrepreneurial intentions of female university students. 2. The variables under study are analyzed from the perspectives of very similar theoretical currents, including the trait theory (TT), the theory of reasoned action (TRA), and the theory of planned behavior (TPB). This research considers exogenous factors (attitudes and behaviors), such as personal traits and psychological profiles, that drive perceived control (perception of business opportunities), which become key elements that lead to reasoned and planned behaviors to raise entrepreneurial intentions in university-going women. Furthermore, this study provides a concrete overview of the importance of business opportunities and psychological profiles as variables that have a mediating influence on the entrepreneurial intentions of university-going women. All of this is considered under the threshold of the post-COVID-19 era, which can lead to new ways of analyzing the conduct and behaviors of individuals who intend to start a business. Likewise, from a methodological perspective, the present study contributes to the literature and empirical studies based on the latest generation of statistical techniques (structural equation modeling with Partial Least Square) by analyzing personal and psychological traits for the detection of opportunities by, and the intentions of, university-going women entrepreneurs. This article contains an introduction, a literary review, and the development of the hypotheses. Following these parts, the methodological section includes the characteristics of the population, the sample, the measurements, and the justification of the variables. Finally, the results obtained, along with relevant discussions, conclusions, implications, and future lines of research, are presented.

## 2. Literature Review and Hypothesis Development

### 2.1. Personality Traits in the Psychological Profile and Entrepreneurial Opportunities

According to Dixit and Moid [48], the personality of the entrepreneur is made up of their psychological characteristics. The psychological well-being of women entrepreneurs could be considered the most important element of their lives; this includes mental and physical health, as well as work performance [49]. Personality traits directly affect an individual’s emotions, thoughts, and behaviors [50]. The personal traits that describe entrepreneurs and that exert an extremely strong influence on behavior are mainly self-efficacy, autonomy, innovation, internal locus of control, achievement motivation, optimism, knowledge, continuous learning, resilience, tolerance to stress, and risk taking, among others [51]. According to Trusić et al. [52], personality traits are key drivers of entrepreneurial behavior and, perhaps, as a dominant factor, more distinctive in entrepreneurs than in other people. The essential trait of a person who becomes an entrepreneur is personal motivation; this is derived from factors that inspire the desire and ambitions of an entrepreneur to maintain interest in and commitment to the required field of work in order to achieve the desired objective [53]. The entrepreneur’s psychological perceptions and cognitions are related to success, confidence, and risk, which have an impact on intentions [13]. Van Scotter and Garg [54] explained that motivation is the physiological will of a person to work continuously with the intention of promoting the new business without abandoning efforts. Having confidence in one’s skills, as well as the knowledge and ability to start a new business, increase entrepreneurial alertness and, thus, lead to the creation of more businesses [55]. In this sense, in order to grow and develop students’ entrepreneurial intentions, there needs to be an effort to improve their personality traits [56]. Derived from the review of the literature, the following approach was constructed.

**Hypothesis** **1** **(H1).***Personality traits (abilities and attitudes) positively influence the strengthening of the psychological profiles (behaviors and emotions) of entrepreneurial university-going women*.

Personality plays an important role in business creation and business success [57,58]. Students with proactive personalities are more motivated to start their companies; this is because they are more capable of exploring the environment in search of opportunities, initiating and undertaking actions, and persisting until they achieve their goals [59]. Zampetakis [60] agrees with these authors that students with proactive personalities, in addition to being able to actively seek and process information, take the initiative to create new opportunities or improve current circumstances. According to Luca et al. [61] and Linfang et al. [62], there is a relationship between the personality trait and entrepreneurial intention and training. They discovered that skills, creativity, and proactive personality are important factors that affect the entrepreneurial intentions of students. The support that the university provides to its students, such as the necessary knowledge, skills, internships, and networking opportunities, are essential to launching a new business initiative [63], in addition to entrepreneurial attitudes, practical entrepreneurship, and business skills [1,64]. Female university entrepreneurs may decide to create a company before truly discovering the opportunity for the specific type of business they want to start [65,66]. Success in business for women entrepreneurs is related to perceiving the existence of opportunities, trusting in their own business capabilities, and knowing other businesspeople who participate in the creation of companies [55], as well as self-assessment of the possibility of success through the knowledge and necessary skills that one possesses [67]. Derived from the review of the literature, the following approach was constructed.

**Hypothesis** **2** **(H2).***Personality traits (abilities and attitudes) influence the perception and detection of greater entrepreneurial opportunities (perceived behavioral control) in university-going women*.

When individuals are certain that events result from their own behavior and resources (locus of control), they have the ability to discover existing opportunities [68]. Starting from the TRA and PBT, attitudes shape the behaviors of individuals. Because behaviors are part of a rational and voluntary process, they can manifest themselves in a negative or positive way [39]. Therefore, the behavioral attitude is determined by subjective probability and subjective desirability. Subjective probability is the probability of the perception of a certain behavior and its subsequent consequence (action). Likewise, subjective desirability is the subject’s desire for a certain action or consequence to occur. Therefore, when there are greater environmental factors, attitudes and behavioral control can be seriously affected. In summary, people in a negative mood, compared to people in a positive mood, tend to evaluate events (such as the consequences of a behavior) more objectively and favorably; they are also more likely to judge favorable events as if they were more likely to occur [32,37]. The psychological factors that characterize the profile of an entrepreneur focus more on intrinsic motivations and emotions (fear, euphoria, pleasure, tension, stress, etc.) that significantly affect the behavior perceived by individuals, and a conflict between these factors and the resulting behavior is called cognitive dissonance [37]. This psychological characteristic is often used to predict entrepreneurship, as individuals are more likely to work harder and persevere in achieving intended results, which, in turn, can help create and maintain a successful business [69]. The behavior of women when owning or managing a business reflects their willingness to take advantage of opportunities and their emotional intelligence to run their own businesses [48]. Buttner and Moore [70] said that through entrepreneurship women seek the opportunity to broaden their knowledge and experience, and the freedom to determine their destiny, the pursuit of challenges, and the opportunity for self-determination are among the most important factors. Students with entrepreneurial intentions are certainly committed to developing independently, showing great intentions in a challenging university environment to achieve good performance, advancement opportunities, and success [56,71]. Escolar-Llamazares et al. [72] explained that entrepreneurial behavior is the product of many influences, and that people who carry out entrepreneurial activity have a psychological profile that predisposes them to act in an entrepreneurial manner and, therefore, differentiates them from others. Derived from the review of the literature, the following approach was constructed.

**Hypothesis** **3** **(H3).***The psychological profile (behaviors and emotions) significantly influences the perception and detection of greater entrepreneurial opportunities (perceived behavioral control) in university-going women*.

### 2.2. Personality Traits, Psychological Profile, and Their Relationship with Entrepreneurial Intentions

Globally, the entrepreneurial intentions of university-going women have become one of the growing topics today, because entrepreneurship is considered a key factor for economic growth and job creation [73]. In accordance with Zhao et al. [23] and Obschonka et al. [74], personality traits, such as risk predisposition, sociability, and openness to experience, appear to influence entry into the business world. Likewise, it has been pointed out that personality traits influence entrepreneurial intention more than situational factors [75]. Entering the business world as a woman involves the challenge of learning to effectively direct the company’s activities in addition to meeting the expectations that are part of the entrepreneurial spirit [76]. Women entrepreneurs who have confidence and leadership and management skills can access new markets [77]; in addition, they create a significant portion of their own businesses, diversify the economic landscape, and are the fastest-growing group of small business owners [78,79]. It is essential to recognize and capture the profile and motivation of women who feel motivated to dedicate themselves to business activity [80]. They are more likely to take calculated risks and develop contingency plans if events do not unfold as planned, and they represent a resource for the market economy [81,82]. There is a need to empower university-going women by providing entrepreneurship education and support, especially in developing countries, where the perspective of female entrepreneurship has not necessarily been widely addressed [73,83]. Business education significantly influences the development of personality traits for future entrepreneurs [68]. Derived from the review of the literature, the following approach was constructed.

**Hypothesis** **4** **(H4).***Personality traits (capabilities and attitudes) significantly influence a greater probability of perception of entrepreneurial intentions (subjective desirability) in university-going women*.

Psychological characteristics are key to stimulating entrepreneurial intention. Entrepreneurial intentions are marked by complex mechanisms, in which behavioral models, attitudes towards entrepreneurship, and entrepreneurial self-efficacy intervene [84]. Adapting teaching and learning to students’ psychological profiles and gender differences allows entrepreneurship programs to boost female entrepreneurship [85]. In examining the psychology and entrepreneurship literature, Dixit and Moid [48] showed that the personality of a businesswoman is made up of her psychological characteristics and is placed at the center of decision making and, therefore, the development of strategies that, ultimately, lead to the destiny of the company. Likewise, the role played by the basic psychological needs of autonomy, competence, and relationship in the formation of the attitudes and intentions of university students in the formation of entrepreneurial intentions has been confirmed [86]. However, psychological factors alone cannot fully explain a woman’s decision to become an entrepreneur; it is important to highlight that there are also economic and cultural conditions, as well as individual factors, such as sociodemographic variables, that influence her decision [52,87]. However, among female entrepreneurs, psychological reasons are considered the main drivers of the intention to start a business [88]; these are often called the inner drive that ignites and sustains behavior to satisfy needs [89]. Derived from the review of the literature, the following approach was constructed.

**Hypothesis** **5** **(H5).***The psychological profile (behaviors and emotions) significantly influences the perception of a greater probability of entrepreneurial intentions (subjective desirability) in university-going women*.

### 2.3. The Detection of Business Opportunities in University-Going Women with Entrepreneurial Intentions

Within the framework of the TBP, the decision to start a business begins with the individual capabilities of each entrepreneur, with this theory being the most used to explain this process [10,90,91]. However, conduct and behavior are factors that can catalyze the detection of business opportunities in female university students. Although traditional schemes and thoughts put greater male entrepreneurship above all else, there are biases and gaps regarding the rise that female entrepreneurship has had in recent decades [92]. Recently, the COVID-19 pandemic has revealed that women entrepreneurs are more vulnerable and have had greater loss of income in marginalized and impoverished regions [93]. Developing and strengthening entrepreneurship requires action. Consequently, this action is preceded by and associated with an intentionally rational decision-making process that can energize conduct and future behavior towards the motivation of intention to undertake entrepreneurship [37,91].

Recent studies have revealed that behaviors are influential in the detection of business opportunities for university entrepreneurs; however, certain factors, such as fear of failure and risk aversion, are those that most affect entrepreneurship [7]. In addition, women, within their family environment and with possible disadvantages with respect to the male gender, have been able to be more resilient and proactive in their intention to start a business during dark scenarios [94,95]. It is also important to note the existence of evidence that demonstrates that university-going women have greater self-efficacy, positive attitudes, greater motivation to detect business opportunities, and higher scores on subjective norms than men, which promote greater entrepreneurial intention [96]. Derived from the review of the literature, the following approach was constructed.

**Hypothesis** **6** **(H6).***Greater perception in the detection of business opportunities (perceived behavioral control) allows for an increasing probability of entrepreneurial intentions (subjective desirability) in university-going women*.

From the review of the literature and the justification of the hypotheses, the following theoretical research model was designed (see Figure 1). Taking reference from Krueger [32], in this research, our model analyzes the attitudes (psychological profile and personal traits) and behavioral control (business opportunities) that lead to entrepreneurship (action), without considering subjective norms, as proposed in the original model.

## 3. Methodology

### 3.1. Population and Sample

This study is quantitative in nature, with an explanatory design. For this purpose, use of the stratified sampling technique was considered because the population is made up of segments and/or categories. The confidence level used to determine the sample was 95%, with a margin of error of 2.6%, and a probability for and against of 50%. Two of the most important universities established in the northwest of Mexico for more than 5 decades participated in this study. The first university is the Autonomous University of Baja California (UABC). Based in the city of Mexicali, it had 7714 students in the areas of Administration, Social Sciences, and Humanities registered during the year 2022 [97]. On the other hand, the Technological Institute of Sonora (ITSON), in its academic units established in the city of Obregón, Guaymas, and Navojoa, had an average of 5450 students studying in the areas of Administration, Social Sciences, and Humanities registered during the year 2022 [98]. The total approximate population is 13,164 students who are between 17 and 30 years of age, of whom 62% are women (see Table 1). The total sample obtained included 1197 university-going women, 60% of whom belong to the UABC and 40% of whom belong to ITSON. To collect the information, a digital questionnaire prepared using Google Forms and sent via email was used. Data were collected during the second half of 2022.

### 3.2. Validation of the Questionnaire

To validate the content of the questionnaire, the theories that supported our study were used, which are those directed at the behavior of individuals towards entrepreneurship, thus focusing our study on the theory of traits and the theory of planned behavior. This phase of our study was carried out through an exhaustive review of the literature to correctly understand and define the dimensions included in the questionnaire and its adaptation to the social context in which this study was conducted. The questionnaire was divided into two phases: the first included questions about general or demographic data (gender, age, degree, and university, among others). The second phase included questions related to personal traits (6 items), psychological profile (4 items), detection of business opportunities (3 items), and entrepreneurial intentions (10 items). To this end, several theoretical and empirical studies related to trait theory and the theory of planned behavior were analyzed. The questionnaire was validated by a panel of experts in university entrepreneurship. The panel was made up of 2 researchers from the UABC in Mexico, 2 researchers from ITSON in Mexico, and 2 researchers from the Polytechnic University of Cartagena in Murcia, Spain. During this review, the panel of experts issued recommendations regarding the design of the dimensions, the wording, the relevance, and the representativeness of each of the questions [99]. Through the content validity index, the group of experts evaluated each of the items of the dimensions on a scale from 1 to 7, in which 1 indicated little relevance and 7 indicated very relevant. A pilot test was also carried out with 50 students who belonged to the population selected for this study. In the literature, there is no total consensus on the percentage or total of the population that should participate in a pilot test [100]. Generally, the subjects who participate in a pilot test are few (a small sample). The information collected prior to the survey, which was applied to the entire sample, allowed us to correct possible errors in the writing, structure, and understanding of the content of the questionnaire items [101]. Subsequently, a factor analysis and Harman’s single-factor test (common method variance (CMV)) were performed [100,102]. This test requires performing the following procedures: (1) running a factor analysis of the exogenous latent and endogenous latent constructs of the investigation, and then an analysis of the principal components without selecting any type of rotation method; and (2) analyzing the values of the non-rotated components and the number of factors that complement the variance [103]. The results of this test were as follows: (1) the model was grouped by 4 factors, (2) the Kaiser–Meyer–Olkin (KMO) indicator was 0.971 and significant at 99%, and (3) the total explained variance of the 4 factors was 82.9%. The first unrotated factor captured only 36.0% of the total variance of the data (4 factors). Therefore, the two underlying assumptions were not met; that is, a new factor did not emerge, and, furthermore, the first factor did not capture most of the variance. The results of the exploratory factor analysis demonstrate that the factor loadings of the 4 factors exceed the value of 0.707, in accordance with what is recommended by Carmines and Zeller [104] and Chin [105] (see Table 2). Therefore, these results suggested that CMV was not a problem in this study. 

Additionally, the correlation matrix is included to demonstrate the non-presence of common method bias. When the value of the correlations of the latent variables is less than 0.9, the presence of CMV is ruled out [100,106]. Therefore, in this study, there is no evidence of the presence of CMV (see Table 3).

To strengthen the CMV analysis, we carried out the correction procedure at the construct level. To achieve this, we incorporated a marker variable (barriers to creativity) into the research model, which has no direct theoretical relationship with the constructs analyzed in this study [107]. The marker variable was measured with 4 items on a scale from 1 to 7 (not important to very important; B1: rejection of changes; B2: fear of using technology; B3: negative influence of culture; and B4: insufficient economic resources). This analysis showed that there are no changes in the structural relationships (path coefficients), as well as showing evidence of the non-presence of significant changes in the value of the R^2^ of the endogenous variables of the research model. Despite the existence of very small but insignificant changes, we can conclude that CMV was not a problem in this study (see Table 4).

### 3.3. Measurement of Variables

In this section, the measurements of the variables of the research model are explained. All variables were measured under the approach of reflective-type constructs in mode A and under the approach of unidimensional type variables. These types of variables have particular characteristics, such as (1) the direction of causality is from the construct to the indicators, (2) the indicators are highly correlated, (3) eliminating an indicator does not alter the meaning of the construct, and (4) these types of measurements are recommended for a model with constructs focused on the analysis of behavioral sciences [108]. The constructs that were used in the research model are described below. The questionnaire items were designed using a 7-point Likert-type scale (1: totally disagree to 7: totally agree). Due to the nature of the constructs and the research focus, the statistical technique based on variance was chosen through the structural equation modeling (SEM) system. For the analysis of this specific study, the Partial Least Square (PLS) method was used. 

#### 3.3.1. Personal Traits (PET)

This construct was measured considering the trait theory, with a focus on entrepreneurship [109]. The construct was made up of the following 6 items: K1. It is important to me to help other colleagues even without knowing them; K2. The well-being of the people around me is important to me; K3. It is important to me to look for new ways to improve my profession; K4. My profession is important to me as an activity that complements my life; K5. It is important to me to update my knowledge to be more efficient in my profession; K6. My professional training, based on values, is important to me. For its design and adaptation, the studies by De Carolis and Saparito [110] and Gibb [111] were considered. The reliability and validity indicators of the construct are as follows: Cronbach’s Alpha = 0.985, factor loading = 0.918 to 0.993, composite reliability (rho_a, rho_c = 0.985), and average variance extracted = 0.919.

#### 3.3.2. Psychological Profile (PSP)

This construct was measured considering the trait theory, with a behavioral approach towards entrepreneurship [109]. The construct was made up of the following 4 items: I1. I like to work daily to always be among the best; I2. I dare to face any situation to achieve my purposes; I3. I analyze mistakes to learn from them; I4. I stick to the saying “I like to be where the action is”. For its design and adaptation, the studies by Shariff and Saud [112] and Mcgee et al. [113] were considered. The reliability and validity indicators of the construct are as follows: Cronbach’s Alpha = 0.963, factor loading = 0.907 to 0.967, composite reliability (rho_a, rho_c = 0.963), and average variance extracted = 0.866.

#### 3.3.3. Business Opportunities (BUO)

This construct was measured considering the theory of traits and planned behavior [26,37,109]. The construct was made up of the following 3 items: E9. I frequently identify ideas that can become new products or services; E10. I generally have ideas that can materialize into profitable companies; E11. I frequently identify opportunities to start new businesses. For its design and adaptation, the studies by Knight [114] and Bolton and Lane [115] were considered. The reliability and validity indicators of the construct are as follows: Cronbach’s Alpha = 0.966, factor loading = 0.933 to 0.975, composite reliability (rho_a, rho_c = 0.966), and average variance extracted = 0.905.

#### 3.3.4. Entrepreneurial Intentions (ENI)

This construct was measured considering the theory of traits and planned behavior as complementary factors that influence the behaviors and decisions of individuals [26,37]. The construct was made up of the following 10 items: C1. I am prepared to do anything to be an entrepreneur; C2. My professional goal is to become an entrepreneur; C3. I am determined to create a company in the future; C4. I have thought very seriously about the possibility of starting a business; C5. I intend to start a company someday; C6. I intend to establish a company within 5 years of my graduation; C7. I am willing to save to invest in my own company; C8. I am interested in knowing about public financing support for entrepreneurship; C9. I am willing to take advantage of business opportunities when they arise; C10. I am interested in working in a company where I can develop my entrepreneurial attitudes. For its design and adaptation, the studies by Antoncic and Hisrich [116], Bolton and Lane [115], and Shinnar et al. [117] were considered. The reliability and validity indicators of the construct are as follows: Cronbach’s Alpha = 0.982, factor loading = 0.863 to 0.959, composite reliability (rho_a, rho_c = 0.983), and average variance extracted = 0.846.

## 4. Results

### 4.1. Measurement Model

Measurement models, external models, and epistemic relations show the relationships between the constructs and the indicators. External relationships are defined by a specific measurement theory of how the latent variables are measured. In the exposed research model, the constructs were estimated in mode A: Correlation weights: (1) The directional arrows go from the construct to the indicators. (2) This mode is traditionally applied to reflective measurement models [105]. For common factor (reflective) models, consistent PLS (PLSc) was ideally applied [118]. In the measurement model, the analysis of the individual reliability of the indicators, the reliability of the constructs (internal consistency), convergent validity, and discriminant validity were considered (AVE) (see Table 5).

#### Discriminant Validity of the Model

To verify the discriminant validity of the model, the suggestions of Fornell and Larcker [119] and Henseler et al. [120], recommending that the amount of variance captured by a construct from its indicators (AVE) should be greater than the variance shared by the construct with other constructs, were considered. The results (diagonal) of the vertical and horizontal AVE show the correlation among the constructs (see Table 6).

Recently, studies about discriminant validity have expressed that the Fornell–Larcker criterion has deficiencies. Therefore, we decided to analyze the heterotrait–monotrait ratio (HTMT), a method that represents the average of the heterotrait–heteromethod correlations in relation to the average of the monotrait–heteromethod correlations [121]. Therefore, the value of HTMT should be between 0.85 and 0.9. [121,122]. Table 7 shows the values of this test; all of the ratios are below 0.9, thereby confirming the existence of discriminant validity in the model.

### 4.2. Structural Model

This section reports the results obtained from the hypotheses of the structural model of this research. To test the hypotheses, PLS-SEM based on variance was used, because the constructs of the model focus on analyzing the behavior and conduct of female entrepreneurs to detect business opportunities and entrepreneurial intentions. Experts in the methodology (SEM) have concluded that this technique is very appropriate and recommended for the analysis of phenomena related to business sciences, marketing, and information technologies [118,123]. For the measurement model and the analysis of the structural model, PLS-SEM was used with the support of the SmartPLS software in version 4.0.9.6 [124]. For the structural analysis of the data, it is necessary to evaluate the following: (1) the magnitude, algebraic sign, and significance of the path coefficients; (2) a one-tailed Student’s *t*-test with (n − 1) degrees of freedom; and (3) confidence intervals (percentile- and bias-corrected). To obtain these indicators, a bootstrap test with 5000 subsamples is required [125].

Table 8 shows the results of model hypotheses H1, H2, H3, and H6. The values of the path coefficients have a positive and significant effect at 99%. Also shown are the standard deviation, the t value (values are greater than 2), the confidence intervals in percentiles, and the corrected bias (no value of 0 is presented in the structural relationships of the analyzed model). These indicators corroborate that all hypotheses have empirical support. H4 and H5 present partial empirical support due to the value of the effect of f^2^ below the allowed range.

To evaluate the quality, relevance, and fit of the model, the values of the adjusted coefficient of determination R^2^ were analyzed, with the following results: PSP= 0.655, BUO= 0.411, and ENI= 0.428. The predictive quality of the model was evaluated through the Q^2^ value (Stone–Geisser test), and this value must be >0 [105]. The values of the independent variables of the model are as follows: PSP = 0.622, BUO = 0.336, and ENI = 0.269. The effect size was also analyzed through f^2^, as follows: H1: (PET) -> (PSP) = 1.90; H2: (PET) -> (BUO) = 0.040; H3: (PSP) -> (BUO) = 0.100; H4: (PET) -> (ENI) = 0.010; H5: (PSP) -> (ENI) = 0.017; and H6: (BUO) -> (ENI) = 0.188. However, despite the existence of a significant effect in H4 and H5, f^2^ values below 0.02 were manifested, demonstrating a very low effect in these structural relationships. According to Cohen [126], the heuristic rules, the values for this indicator, are as follows: small effect (f^2^ > 0.02); moderate effect (f^2^ > 0.15); and large effect (f^2^ ≥ 0.35). Although there is uncertainty in the use of indicators to measure the global fit of the model, for our study, we have included the following indicators: SRMR, Exact fit criteria d_ULS, d_G, NFI, and Chi² [127]. The standardized root mean square residual (SRMR) is recommended to be <0.08 [128,129]. Our result was 0.041. The values (d_ULS and d_G) were also reported, using the bootstrap-based test for the exact overall fit of the model, and its original value was compared to the confidence interval created from the sampling distribution. The confidence interval must include the original value. Therefore, the upper limit of the confidence interval should be greater than the original value of the fit criteria d_ULS and d_G to indicate that the model has a “good fit”. The confidence interval is chosen so that the upper limit is at the 95% or 99% point [130]. The NFI is defined as 1 minus the Chi² value of the proposed model divided by the Chi² values of the null model. Consequently, the NFI results in values between 0 and 1. The closer the NFI is to 1, the better the fit. It is recommended that the normalized fit index (NFI) value be close to 0.9, and our result was 0.922 (see Table 9). According to the data provided in the estimated model, the variables that build the proposed theoretical model of this research provide compelling evidence of the existence of acceptable quality and predictive relevance, and it fits the theory.

### 4.3. Moderating Effect

This section analyzes the moderating effect that business opportunities exert on the relationship between the psychological profile and personal traits with the entrepreneurial intentions of university students. The literature has exposed different models and perspectives, such as those by Shapero and Sokol [31] and Ajzen [39], which do not completely coincide but argue for the existence of different internal and external factors that intervene in the relationships among attitude, behavior, and intention. Therefore, the mix of attitudes and personal skills generates opportunities that lead to planned behavior and action (intentions). Krueger [32] stated that the combination of the TPB model and the Business Event, along with the perception of business opportunities (desirability and perceived viability), are the antecedents of exogenous factors, such as personal desirability (attitudes and self-efficacy). In summary, the perception of opportunities plays a moderating role between the personal and psychological factors that lead to entrepreneurial intentions.

The analyses presented in Table 10 reveal in H7 the existence of a moderating effect (very weak “f^2^= 0.019”) explaining that when the perception of business opportunities decreases, the intensity of the relationship between the psychological profile and entrepreneurial intentions increases. Furthermore, in H8, the results reveal that the perception of business opportunities does not show any moderating effect on the intensity of the relationship between personal traits and entrepreneurial intentions of university students.

### 4.4. Mediation Analysis

A multiple measurement analysis was included in the research model to verify the indirect effects (c’) of the variables involved in the direct relationships. Following the recommendations of Hayes [131], to carry out this type of analysis, it is necessary to (1) calculate the value of the direct effect (c’); (2) estimate the indirect effects (a1 × b1 + a2 × b2) using the bootstrapping technique, with 5000 subsamples with 90% confidence intervals [132]; and (3) determine the magnitude of the indirect effect through the value of the variance accounted for (VAF), and this indicator must be in a range between 20% and 80%. It is also important to analyze the relevance of the effect to determine the type of mediation [133]. Indirect effects are significant when the value of (0) is not included in the confidence intervals [134,135].

The following hypotheses were generated to analyze the mediation effects: H1: Personal traits (PET) have a direct positive effect on the entrepreneurial intentions (ENI) of university students; H1 = PET → ENI = (c’). H2: The perception of business opportunities (BUO) is a mediating variable that positively influences the relationship between PET and ENI in university students; H2 = PET→BUO→ ENI. H3: The psychological profile (PSP) is a mediating variable that positively influences the relationship between personal traits (PET) and entrepreneurial intentions in university students (ENI); H3 = PET→PSP→ ENI. The results of this mediation test reveal that PET has a positive (moderate to low) and significant direct effect on IPD (H1: c’ = 0.131 ***). H2 reveals that the mediating variable BUO has a positive and significant moderate effect on the relationship between PET and ENI (H2: a_1_ × b_1_= 0.254 ***). H3 explains that the PSP variable manifests a positive (moderate to low), indirect effect on the relationship between PET and ENI (H3: a_2_ × b_2_= 0.144 ***). This analysis explains that the total indirect effect is 0.398 ***, and the total effect is 0.529 *** (See Table 11 and Figure 2). The value of the magnitude of the indirect effect is 75.0% according to the VAF value obtained. With these results, this analysis demonstrates the existence of complementary partial mediation because all of the hypotheses in this model have the same (positive) direction, and the VAF value is within the allowed parameters. In short, it is understood that the measuring variable that most influences the relationship between PET and ENI is the perception of business opportunities (BUO); however, when the BUO and the PSP are combined, they exert a strong indirect effect on the results of the relationship between personal traits and entrepreneurial intentions of university students. Despite the divergence in the literature regarding the correct measurement of entrepreneurial intention, the main theoretical currents reveal that there are many factors that influence human intentions and decisions to achieve entrepreneurship. Based on Krueger’s premises, the perception of business opportunities is the link among attitudes, behaviors, intentions, and actions [32]. Furthermore, the process of identifying business opportunities occurs over time, rather than just in a single moment of inspiration. Opportunity identification is the result of personal, social, cultural, and technical forces that lead to the perception of a possible business opportunity [136]. Some empirical studies have researched factors that lead to the detection of opportunities as a key element in trying to start a business. For example, Rosique-Blasco et al. [137] and Villanueva-Flores et al. [138] discovered that perceived behavioral control (perception of business opportunities) and subjective norms mediate the relationship between psychological capital and entrepreneurial intention. On the other hand, studies by Ouni and Boujelbene [19] and Otache et al. [139] showed that self-efficacy and alertness are included in the capabilities that an individual develops to detect business opportunities, and that, in turn, they are variables that mediate the relationships among attitudes, subjective norms, and perceived control with business intentions.

## 5. Discussion

Within the context of the complementary theories used in this study (TT, TRA, and TPB), the findings derived from the hypothesis tests of the research model are discussed below. Our model analyzes attitudes, divided into personal traits and psychological profile (capabilities, behaviors, and emotions), that affect business opportunities (perception of perceived control) and entrepreneurial intentions (subjective desirability). The variable (subjective norms) was not included in this model, with the purpose of validating only the effect of exogenous environmental factors that influence the business behaviors of women entrepreneurs.

In one scenario, the results of H1 explained that the personal traits possessed by university-going women strongly influence the consolidation of the psychological profile. The most significant personal traits were the following: K3. It is important to me to look for new ways to improve my profession; K5. It is important to me to update my knowledge to be more efficient in my profession; and K6. My professional training based on values is important to me. With this, it was corroborated that knowledge, values, and self-efficacy are intrinsic factors that are most developed in female university students. The TPB explains that the three components of this theory influence behavioral intentions (emotions and behaviors), including personal traits of behavior and its results, which is called the attitude toward the behavior. On the other hand, subjective norms, such as social pressure towards a behavior, are factors that hinder and distort behavior [37,109]. Confirming this, personal traits, such as cognitive and emotional mechanisms, affect the ability of an entrepreneur, when making personal judgments, to make decisions and express their feelings [140,141]. These findings are in line with previous empirical studies analyzed explaining that personality traits directly affect the emotions (achievement and task motivation), thoughts, and behaviors of an individual towards entrepreneurship [50,52,53].

In a second scenario, we analyzed how the personal traits (attitudes and abilities) and the psychological profile (behavior and emotions) of university-going women affect the perception of greater business opportunities. Hypotheses H2 and H3 reveal that personal traits have an influence on business opportunities and entrepreneurial intentions. Hypothesis H2 emphasizes, without a doubt, that self-efficacy, perseverance, sociability, motivation, and proactivity are the traits that stand out in female university students in this region, given that they are drivers towards the detection of business opportunities. In addition, women entrepreneurs are more likely to trust their own abilities and skills to interact with established businesspeople with a view to starting a business [96]. These findings are consolidated and aligned with the TBP, revealing that social paradigms are frequently broken by the behaviors of entrepreneurial women and female resilience during current times [37,142]. Previous studies highlight and support our findings, explaining that women have greater self-efficacy and confidence when exploring business ideas to put them into action [65,143]. Therefore, TT and TPB explain how attitudes, such as motivation, skill, and creativity, are drivers that shape psychological traits and emotions for decision making and the detection of business opportunities [91,144]. Likewise, previous studies have pointed out that locus control [68,137] and emotional intelligence in women entrepreneurs enable them to be alert and detect greater opportunities to do business than men [69,145]. Hypothesis H3 reveals that personal traits strongly influence the detection of business opportunities by university-going women. The TT has explained that personal traits can be positive (e.g., proactivity, commitment, self-efficacy, and motivation) [91] or dark (e.g., narcissism, psychopathy, Machiavellianism, and sadism) [26]. However, in our study, we considered the positive personal traits that encourage and fuel entrepreneurial alertness (detection of business opportunities and the intention to start a new business). Our findings are aligned with previous empirical studies demonstrating that female entrepreneurship in the last two decades has had a significant boom for the economies of different countries [73]; this is due to their leadership capacity, self-efficacy, and management capacity in adverse environments [77,79].

In a third scenario, we analyzed how the personal traits (attitudes and abilities) and the psychological profile (behavior and emotions) of university-going women affect the probability of increasing entrepreneurial intentions. Hypothesis H4 yields significant data according to the value of the path coefficient, the t value, and its level of significance. However, we can also observe that the value of the f square is below 0.02, which may cause the effect generated by personal traits to slightly influence business intentions. With these findings, it can be verified and/or inferred that there are other elements of the environment, such as social norms, that, at this time, may be more strongly affecting behaviors towards entrepreneurial intentions in university-going women [146]. The behaviors and qualities of the psychological personality that stand out most in university-going women are the following: I1. I like to work daily to always be among the best and I3. I analyze mistakes to learn from them. Despite the increase in entrepreneurship developed by women, there is still evidence of strengthening some personal traits that further motivate and awaken entrepreneurial alertness to increase the control of behavioral perceptions towards action (entrepreneurial intentions) [76,80]. It is likely that the fear of failure, insecurity, low self-efficacy, and low self-esteem are affecting these behaviors, in addition to strong economic effects and social norms, which have been derived from the havoc wreaked by the COVID-19 pandemic [147].

On the other hand, Hypothesis H5 shows that the psychological profile has a greater effect than personal traits on triggering the entrepreneurial intention of university-going women. However, like H4, the value of the f square is below the allowed value. This can be interpreted as a relatively low ratio that depends on other internal and external factors. In short, the postulates of the TBP indicate that there must be a balance among attitudes, subjective norms, and individual emotions in order to have greater control of behavior towards entrepreneurial intention [37,148]. However, our findings are partially in the same direction as previous studies and theoretical currents (TRA and TBP) in arguing that personal traits and psychological profile may be affecting behavior due to other underlying factors when processing opportunities and entrepreneurial intentions [32,86,88].

The fourth scenario that we analyzed refers to hypothesis H6, proving that the direction of business opportunities strongly influences the entrepreneurial intentions of university-going women. These findings show that TT and TPB are complementary theories that analyze personal traits, behavior, attitudes, and perceived behavioral control towards entrepreneurship. These theories are complementary (they can also influence bidirectionally), given that personal traits significantly influence the behavior, emotions, subjective norms, and entrepreneurial behavior of individuals [149]. The results presented in our study are in the same direction as those of the previous studies analyzed. Previous research shows that proactivity, risk taking, and an innovative attitude are drivers for detecting business opportunities and comprise a perfect means that calls for action to undertake. It is also highlighted that, recently, female entrepreneurship has shown better results due to the capacity for adaptation, leadership, and resilience of female entrepreneurs in the face of scenarios plagued by social and financial uncertainty [7,94,96].

In the fifth and final scenario, we analyzed the results derived from the moderation and mediation effects that were executed in the research model. The results revealed that the detection of business opportunities has a negative moderating effect when the psychological profile is related to entrepreneurial intentions. In other words, as the perception of business opportunities fades, the intensity between PSP and ENI decreases. On the other hand, no empirical evidence was found to prove that business opportunities affect the relationship between PET and ENI. Furthermore, the mediation analysis revealed that the combination of business opportunities and psychological profile exerts a substantial and significant indirect effect on the relationship between the personal traits and entrepreneurial intentions of university-going women. However, where the greatest positive impact occurs is when the BUO variable intervenes in the relationship between PET and ENI. These results are consistent with previous studies, as well as with the theory of reasoned action and the TPB [32,37].

## 6. Conclusions

In short, this research has answered the objectives and questions posed in our theoretical model, which focused on the following: 1. Do personality traits influence the psychological profiles of female university students? 2. Are personal traits and psychological profiles contributing attitudes that influence behavior towards the perception of opportunity detection and entrepreneurial intentions in female university students? and 3. Do the detection of opportunities and the psychological profile have moderating and mediating effects on the entrepreneurial intentions of female university students? These questions have been amply explained in our research and by the existing literature. It was revealed that personal factors and the psychological profile are attitudes that positively affect behavior towards greater perception of the detection of business opportunities; however, they require other internal and/or external factors that can play the role of measurers between attitudes and action towards the desirable behavior of entrepreneurial intention [32].

Our contributions to the literature with this research are noteworthy because the findings have explained that the personal traits and psychological profiles of university-going women are factors that determine the detection of business opportunities and entrepreneurial intentions. From the theoretical context, this study brings to light that there is an important window to continue analyzing and further strengthening the internal and external factors that affect female entrepreneurship from the perspectives of trait theory and planned behavior. Although there is a boom in the literature related to research on female university entrepreneurship, there are still important gaps, and gaps that encourage us to continue discovering more relevant data about this phenomenon [64,150]. From a Latin American context, this study can be a reference and generalizable model based on the behaviors of university-going women towards the intention of entrepreneurship. Social, cultural, economic, and health changes, such as the COVID-19 pandemic, have been barriers that have limited entrepreneurial development for both men and women. This is more aggravated in the countries of the Latin American region, which are also going through difficulties related to investment and production, the global financial crisis, geopolitical tensions, war, and the resurgence of inflation [44]. This profound scenario of permanent, multidimensional crisis with an unequal recovery has had a greater impact on women. The pandemic has aggravated the persistent structural knots of gender inequality. Where this crisis deepens most is in socioeconomic inequality and poverty, as well as in discriminatory and violent patriarchal cultural patterns, the culture of privilege, the coverage of education, the sexual division of labor, and the concentration of power [151]. 

Particularly for Mexico, strengthening and raising entrepreneurial intentions in the female context represents an important key to economic growth, as well as social and cultural development. With entrepreneurial education not only in universities, but at all educational levels, the lack of job opportunities, unemployment, underemployment, and business informality can be combated. However, historically, Mexican society is not characterized by cultivating a culture towards entrepreneurship, which represents an enormous job for educational institutions and for the entrepreneurial university. Data from the latest population and housing census show that Mexico currently has more than 126 million inhabitants, of which 52% are women and 48% are men [152]. In addition, there are, on average, 25 million women between the ages of 6 and 25, of which 60% have access to education [153]. However, economic problems, inequality, poverty, and, recently, the COVID-19 pandemic have affected the terminal efficiency index at all levels of the country. Despite this, these data reflect enormous potential and a focus on improving educational opportunities in Mexico so that, in turn, better and new public policies are carried out that affect culture and education towards entrepreneurship. All of this is in favor of the social, cultural, and economic well-being of a region classified as under development for many years.

From a practical perspective, our study also contributes to the development of strategies and initiatives by interested parties; therefore, this study suggests the following implications: (1) it is important for universities to detect the motivations, capabilities, and skills (hard and soft) that contribute to the strengthening of teaching practices focused on alertness and entrepreneurial intention [154,155]; (2) university managers should seek and adapt new educational models, as well as the methodologies of successful business incubators and accelerators (North American and European) to promote university entrepreneurship [156,157]; (3) university managers should adopt and strengthen gender inclusion models for the development of innovative and technological entrepreneurship [10,158]; (4) it is important to develop and promote public policies focused on inclusion, equity, and discrimination to raise the quality of business opportunities and university entrepreneurship [159]. In these initiatives, universities, governments, business sectors, and social opinion leaders will have to be linked and articulated [160]. Finally, the fifth recommendation or implication refers to the adoption of training programs and emotional counseling for university students, given the serious problems that arise in these new generations of the population [161].

As in all research, our study is not free of limitations; therefore, in this section, we describe those that are the most significant. The first limitation refers to the design of the questionnaire; the instrument to collect data is built from adaptations of other studies in other sociodemographic contexts. The second limitation is related to the analysis of unidimensional constructs. In future research, we may consider multidimensional constructs; however, the vast majority of these can be analyzed both ways. The third limitation refers to the statistical technique used, which focuses on the analysis of variance. In the future, covariance analysis can be considered, with models based on CB-SEM. Finally, another limitation is the sample, given that the participants come from two universities. However, these educational institutions are very representative of the northwest of Mexico. In the future, participants from other regions and other countries may be considered in order to compare the results. The study of female entrepreneurship has awakened and generated greater interest among researchers in administrative sciences, psychology, and sociology in a very notable way; however, it is important to continue with these types of studies to monitor phenomena related to personal traits, attitudes, subjective norms, and behaviors towards entrepreneurial intentions in the university context. Therefore, we suggest adding new variables that consider social, cultural, economic, and environmental aspects. Understanding that behavior and conduct towards entrepreneurship are dynamic attitudes, it is important to consider more in-depth analysis that leads to the continuous and specific analysis of this phenomenon in order to identify the causes that lead entrepreneurs to put into practice the detection of opportunities and entrepreneurial intention. For future work related to female entrepreneurship, it is important to add the internal and external factors that influence these entrepreneurial behaviors and intentions, such as the family environment, the educational context, and the government environment.

## Figures and Tables

**Figure 1 behavsci-14-00066-f001:**
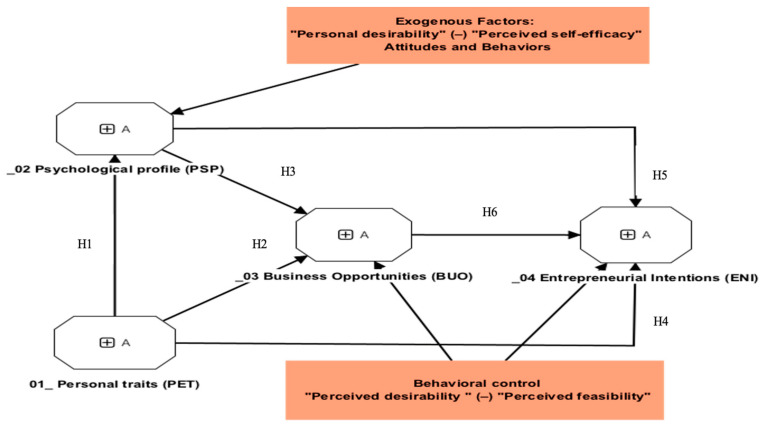
Theoretical operating research model.

**Figure 2 behavsci-14-00066-f002:**
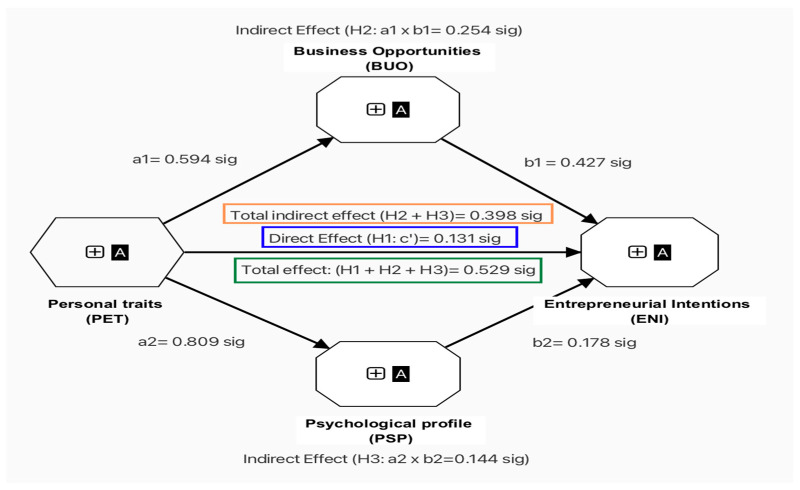
Mediation analysis of the research model.

**Table 1 behavsci-14-00066-t001:** Sample characteristics.

Age	UABC	ITSON	Total	%
17–20	443	296	739	62%
21–25	250	167	417	35%
26–30	25	16	41	3%
Total	718	479	1197	100%

**Table 2 behavsci-14-00066-t002:** Factor loadings of the model components.

Factor 1 (PET)	Factor Loading	Factor 2 (PSP)	Factor Loading	Factor 3 (BUO)	Factor Loading	Factor 4 (ENI)	Factor Loading
K1	0.918 ***	I1	0.907 ***	E10	0.945 ***	C1	0.863 ***
K2	0.944 ***	I2	0.933 ***	E11	0.933 ***	C2	0.907 ***
K3	0.993 ***	I3	0.967 ***	E9	0.975 ***	C3	0.912 ***
K4	0.959 ***	I4	0.914 ***			C4	0.944 ***
K5	0.966 ***					C5	0.935 ***
K6	0.969 ***					C6	0.868 ***
						C7	0.928 ***
						C8	0.940 ***
						C9	0.959 ***
						C9	0.959 ***
						c10	0.936 ***

Note: Personal Traits (PET), Psychological Profile (PSP), Business Opportunities (BUO), Entrepreneurial Intentions (ENI); *** *p* < 0.001.

**Table 3 behavsci-14-00066-t003:** Correlation matrix.

Variables	(PET)	(PSP)	(BUO)	(ENI)
(PET)	1.000	0.809	0.594	0.528
(PSP)	0.809	1.000	0.623	0.550
(BUO)	0.594	0.623	1.000	0.615
(ENI)	0.528	0.550	0.615	1.000

**Table 4 behavsci-14-00066-t004:** CMV analysis with marker variables.

Construct Relationships (without Marker Variable)	Path Coefficient	SD	T Score	Construct Relationships (with Marker Variable)	Path Coefficient	SD	T Score
(PET) -> (PSP)	0.809 ***	0.019	41,987	(PET) -> (PSP)	0.802 ***	0.020	40,507
(PET) -> (BUO)	0.260 ***	0.054	4819	(PET) -> (BUO)	0.256 ***	0.054	4778
(PSP) -> (BUO)	0.413 ***	0.056	7365	(PSP) -> (BUO)	0.409 ***	0.056	7278
(PET) -> (ENI)	0.131 ***	0.047	2801	(PET) -> (ENI)	0.125 ***	0.046	2624
(PSP) -> (ENI)	0.177 ***	0.051	3446	(PSP) -> (ENI)	0.165 ***	0.050	3306
(BUO) -> (ENI)	0.427 ***	0.041	10,394	(BUO) -> (ENI)	0.420 ***	0.040	10,596
R^2^ (PSP)	0.655 ***			R^2^ (PSP)	0.656 ***		
R^2^ (BUO)	0.411 ***			R^2^ (BUO)	0.412 ***		
R^2^ (ENI)	0.428 ***			R^2^ (ENI)	0.448 ***		

Note: *** *p* < 0.001; SD (standard deviation).

**Table 5 behavsci-14-00066-t005:** Internal consistency and convergent validity.

Construct	Cronbach’s Alpha	Composite Reliability (rho_a)	Composite Reliability (rho_c)	Average Variance Extracted (AVE)
Personal Traits (PET)	0.985	0.986	0.985	0.919
Psychological Profile (PSP)	0.963	0.963	0.963	0.866
Business Opportunities (BUO)	0.966	0.967	0.966	0.905
Entrepreneurial Intentions (ENI)	0.982	0.983	0.982	0.846

**Table 6 behavsci-14-00066-t006:** Discriminant validity (Fornell and Larcker).

Construct	AVE	(PET)	(PSP)	(BUO)	(ENI)
Personal Traits (PET)	0.911	**0.959**			
Psychological Profile (PSP)	0.869	0.809	**0.931**		
Business Opportunities (BUO)	0.898	0.594	0.623	**0.951**	
Entrepreneurial Intentions (ENI)	0.855	0.528	0.550	0.615	**0.920**

**Table 7 behavsci-14-00066-t007:** Discriminant validity (Heterotrait–Monotrait Ratio (HTMT)).

Construct	AVE	(PET)	(PSP)	(BUO)	(ENI)
Personal Traits (PET)	0.911				
Psychological Profile (PSP)	0.869	0.809			
Business Opportunities (BUO)	0.898	0.594	0.623		
Entrepreneurial Intentions (ENI)	0.855	0.528	0.549	0.616	

**Table 8 behavsci-14-00066-t008:** Model hypothesis testing.

Hypothesis	Path Coefficient	SD	T Score	CI		Bias Corrected 5% CI	95% CI	% Explained Variance	Result
				5%	95%				
H1: (PET) -> (PSP)	0.809 ***	0.019	41.987	0.775	0.839	0.775	0.839	65.4%	Supported
H2: (PET) -> (BUO)	0.260 ***	0.054	4.819	0.173	0.351	0.173	0.351	15.4%	Supported
H3: (PSP) -> (BUO)	0.413 ***	0.056	7.365	0.320	0.505	0.320	0.505	25.7%	Supported
H4: (PET) -> (ENI)	0.131 ***	0.047	2.801	0.058	0.212	0.058	0.212	6.9%	Partial support
H5: (PSP) -> (ENI)	0.177 ***	0.051	3.446	0.089	0.257	0.089	0.257	9.7%	Partial support
H6: (BUO) -> (ENI)	0.427 ***	0.041	10.394	0.359	0.494	0.359	0.494	26.3%	Supported

Note that n = 5000 subsamples; *** *p* < 0.001 (one-tailed Student’s *t*-test): t (0.05; 4999) = 1645; t (0.01; 4999) = 2327; t (0.001; 4999) = 3092.

**Table 9 behavsci-14-00066-t009:** Model fit.

	Saturated Model	Estimated Model
SRMR	0.022	0.041
d_ULS	0.134	0.461
d_G	0.563	0.654
Chi^2^	3727.880	4235.760
NFI	0.920	0.922

**Table 10 behavsci-14-00066-t010:** Moderating effects.

Hypothesis	Path Coefficient	SD	T Score	f^2^	*p* Value
H7: (BUO) × (PSP) -> (ENI)	−0.048 ***	0.020	2.425	0.019	0.015
H8: (BUO) × (PET) -> (ENI)	−0.023 ^nsig^	0.021	1.086	0.001	0.278

Note that n = 5000 subsamples; *** *p* < 0.001 (two-tailed Student’s *t*-test): t (0.01; 4999) = 1645; t (0.05; 4999) = 2577; t (0.001; 4999) = 3.292. ^nsig^ = not significant.

**Table 11 behavsci-14-00066-t011:** Mediation analysis.

Hypothesis (Effects)		Bootstrap 90% (CI)		Bootstrap 90% (CI)		
Direct Effect	Coefficients	Percentiles	Bias Corrected			
H_1_: c’	0.131 ^sig^	0.056	0.209	0.055	0.054	
a_1_	0.594 ^sig^	0.549	0.639	0.549	0.548	
a_2_	0.809 ^sig^	0.777	0.776	0.776	0.775	
b_1_	0.427 ^sig^	0.358	0.493	0.359	0.359	
b_2_	0.178 ^sig^	0.094	0.261	0.094	0.094	
Indirect Effect	Point Estimate	Percentiles	Bias Corrected			VAF
H_2_: a_1_ × b_1_	0.254 ^sig^	0.196	0.315	0.197	0.197	
H_3_: a_2_ × b_2_	0.144 ^sig^	0.073	0.219	0.073	0.073	
Total indirect effect (H_2_ + H_3_)	0.398 ^sig^	0.270	0.534	0.270	0.270	75.0%
Total effect: (H_1_ + H_2_ + H_3_)	0.529 ^sig^					

^sig^ = *p*-value of 99% significance. Bootstrap 90% CI (confidence interval).

## Data Availability

The data and the questionnaire used in this study are available to other authors who require access to this material.

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
