# Peer review of "Personal and Psychological Traits of University-Going Women That Affect Opportunities and Entrepreneurial Intentions"

_behavsci, 2024, doi:10.3390/bs14010066_

Round 1

Reviewer 1 Report

Comments and Suggestions for Authors

Dear Author(s):

I had the opportunity to review your paper entitled "Personal and Psychological Traits of University Women that Affect Opportunities and Entrepreneurial Intentions" submitted to the Journal of Behavioral Sciences. In this quantitative study, you gathered survey data from female students in Mexico using Partial Least Squares Structural Equation Modeling (PLS-SEM). Overall, your study is highly flawed and shows an inacceptable lack of scientific rigor for publication. Please understand my review as a potential learning experience to avoid desk-rejection in the future. Hopefully, my remarks will help you to improve the way how you conduct research. Best of luck!

1.     First, the English grammar is very bad. You rely on passive voice mostly and use weird sentence structure (without any verbs sometimes or line 168: “X, this researcher agrees with” – why not “X agrees with”?) and it seems you do not care about proper punctuation at all. Often you contradict yourself (e.g., the dark “triad” consists of three not six factors: triad=3, etc.), making it hard to follow you line of argumentation. I could give you at least 30 grammar-related examples, but this will not help because you are not English natives. Therefore, I highly recommend you use a professional proofreading service. Worst of all, you even have some sentences in Spanish (e.g., 484-486; 504-506). Overall, this leaves a very bad impression and low attention to detail on your end. Do your best to avoid such mistakes.

2.     Your theoretical positioning is all over the place. What is missing is a big why. Why do we need your study? Why should we care about female entrepreneurs in Mexico? Please better motivate your study in the introduction. Insufficient research on a topic or controversies or disagreements may not be strong-enough reasons to motivate a study. What would happen if we did not answer your research questions? What do we know already about the research question as it applies to theory and practice? And what do we not yet know, but urgently must learn? How does your study fill a significant gap in theory (and practice)? What are your key contributions? You should aim for easy-to-digest takeaways in the study’s conclusions supported by the data.

3.     Theoretically, you mention trait theory and the theory of planned behavior (TPB) without further explanations of underlying assumptions. On top, the TPB predicts behavior based on intentions. However, the essential entrepreneurial behavior is not part of your hypothesized model. Moreover, your model suggests several mediation hypotheses without you ever testing for mediation or conditional indirect effects (Hayes, 2018). Obviously, business opportunities (BUO) arise exogenously more often than endogenously: the environment presents us with opportunities for us to discover and realize the value. Having said this, BUO should be a moderator affecting the relationship strengths between internal traits and the manifestation of entrepreneurial intentions (ENI). Specifically, in the way you suggest the causal chain, BUO drives ENI. But should an individual first develop an intention to go for it an discover opportunities. In your logic, opportunities arise randomly, further cementing them to be exogenous rather than determined by personality traits. This relates to the chicken-and-egg dilemma: BUO leads to ENI or vice versa? With cross-sectional data and your findings suggesting partial mediations only, you cannot rule out either direction, hence require even more theoretical support.

4.     Your arguments do not fit your hypothesized model. You argue about many psychological and personality traits that you fail to cover in your measurement models. Please pose your hypotheses AFTER the theory deduction and not before to avoid confusion. Also, most hypotheses lack a clear direction, either positive or negative. What does “X influences/explains Y” imply? Without a relationship direction, these hypotheses make no sense. Similarly, the construct labels (e.g., PET or PSP) could be any trait or profile. You seem to measure different traits at the item-level and aggregate them reflectively at the construct level, equivalent to a higher-order construct. This approach makes your hypotheses even more abstract and nonsensical.

5.     Surprisingly, despite this lack of theoretical focus, your reliabilities are close to one, suggesting a severe issue with your data. Looking into the item wordings, most these items are heavily socially desirable, further fueling common method bias and reducing discriminant validity between constructs. The HTMT threshold should be 0.9, sometimes 0.85, not 1! Do you confuse this threshold with Fornell-Larcker maybe? Why are Table 3 and Table 4 values almost identical? – There should be differences!  

6.     Methodologically, there are to many mistakes for me to address all of them, so just some examples: Confidence intervals cannot exclude the estimated mean path coefficient. In your case, even the upper 95% estimate is below the path coefficient (cf. Table 5). Two effect sizes are meaningless (H3 & H5: f²<.02). Still, you fail to reject them for being statistically significant because of the sheer statistical power but not important (f²<.02). Table 6 shows same values for saturated and estimated models, how is this possible? Reported thresholds are wrong: SRMR should be “<” and not “>.08.” You should provide the latent correlation matrix to understand total vs. conditional effects and potential mediation.

7.     In general, when it comes to subjectively perceived data, constant culturally induced method variance (CMV) typically experienced with highly collectivistic cultures (e.g., conformity bias, social desirability) will not vanish. Please use state-of-the-art CMV testing via the comprehensive CFA marker technique by Williams et al. (2010). This tool is very powerful by providing explicit diagnostics where common method variance occurs, on the measurement or structural level, or even both. I am pretty sure most of your items show mean values above the theoretical mean of 4, probably closer to 6 than 4. What are the minimum values, did respondents use to full range of the scales from 1 to 7? Overall, social desirability is a huge threat to external validity and probably drives most of your findings.

8.     Contrary to your statements, PLS is a SEM method (line 353). How can you define 5% of the sample for the pilot study prior to collecting the data (line 326)? Do you mean the population instead of sample? Stratification considers representativeness, not precision (line 288)! How can the first extracted factor be more than the total explained variance. It is always less than the total variance, so your statement is redundant (line 337). You probably mean less than 50%? Anyways, there are many more wrong statements, and I don’t see how you can improve to a point where the quality is sufficient for publication.

9.     Minor issues: Several typos, e.g., line 409: Chin, 1998, not Chinn, 1998. Line 368: PSP, not PEP.

10.  Your discussion is irrelevant at this point. However, I cannot identify truly novel ideas not already discussed in past literature. You could connect better with past findings and contribute to a consensus on this matter.

In conclusion, your paper requires a total overhaul. I hope my assessment is not discouraging but helpful to further learn and improve your methodological skills. All the best!

References

Hayes, A. F. (2018). Introduction to mediation, moderation, and conditional process analysis: A regression-based approach. Guilford publications.

Williams, L. J., Hartman, N., & Cavazotte, F. (2010). Method variance and marker variables: A review and comprehensive CFA marker technique. Organizational research methods13(3), 477-514.

Comments on the Quality of English Language

Very bad grammar!

Author Response

Thank you very much for your comments, suggestions and observations. We have tried to answer all of his questions; we are sure that his theoretical and methodological contribution will contribute strongly to this research.

Regards

Reviewer 2 Report

Comments and Suggestions for Authors

The issue being addressed is interetsing but the paper needs a lot more work before it can eb ready.

1. The abstarct is not informative need to be rewritten.

2. The justification for research is not well develloped, ie; why still study intention?

3. The theory is not correct as TPB should have Attitude, Subjevtive norms and Percevied behavioral control and Intention but this study only has Intention.

4. One variable is not a theory as theory is a combiantion of variables put together to explain a phenomenon.

5. Review of the literature needs to be updated.

6. The review should also tie the variables being hypothesized theoretically.

7. Instrument sources should be clearly stated as I think the items are not measuring the variables in the model.

8. Analysis has a problem of duscriminant validity as the HTMT ratios are very high with one over 0.90 which points to serious problem of discriminant validity.

9. Discussions need to compare and contrast with the literature.

10. Implications need to be implementable.

Comments on the Quality of English Language

Acceptable.

Author Response

Thank you very much for your comments, suggestions and observations. We have tried to answer all of his questions; we are sure that his theoretical and methodological contribution will contribute strongly to this research.

The abstract is not informative need to be rewritten.

  1. Answer: Thank you for your suggestion, we have rewritten the abstract in order to more precisely explain the objective, the subjects under study and the main findings of this research. In addition, more results derived from the moderation and mediation analyzes were incorporated. You can verify the changes in the final version of the manuscript.

The justification for research is not well develloped, ie; why still study intention?

  1. Answer: Thank you very much for your suggestion and observation, we have added greater theoretical support in the introduction section and clarified more precisely the importance and contribution of our study. You can verify the changes in the final version of the manuscript.

The theory is not correct as TPB should have Attitude, Subjevtive norms and Percevied behavioral control and Intention but this study only has Intention.

  1. Answer: Thank you very much for your suggestion and observation, we have added greater theoretical support in the literature review and justification of the hypotheses section, with the purpose of improving the hypothetical relationships of the model. It is also important to clarify that our model contemplates attitudes and behavior towards business opportunities and towards entrepreneurship (action). Leaving aside the social norms of the original model proposed by Ajzen (1991), it has also been placed and linked to the theory of reasoned action as the theoretical background. Therefore, the research contemplates the analysis of exogenous factors (personal traits and psychological profile), as determining elements that predict the perceived behavioral behavior, which makes our model more interesting and different from previously carried out studies. You can verify the changes in the final version of the manuscript.

One variable is not a theory as theory is a combiantion of variables put together to explain a phenomenon.

  1. Answer: Thank you very much for your suggestion and observation, we agree with your comment, the proposed research model includes 4 variables which are part of the trait theory, the theory of reasoned action and the theory of planned behavior. Therefore, this study considers these theories as complementary to analyze the theoretical model from these perspectives. You can verify the changes in the final version of the manuscript.

Review of the literature needs to be updated.

  1. Answer: Thank you very much for your suggestion and observation. In the introduction and justification of the hypotheses section we have added more information that allows us to visualize greater theoretical solidity and consistency. Although, there are classic authors that we have not been able to eliminate due to the value they contribute to the research. You can verify the information in the final version of the manuscript.

The review should also tie the variables being hypothesized theoretically.

  1. Answer: Thank you very much for your suggestion and observation. In the literature review and justification of the hypotheses section we have added more information that allows us to visualize greater theoretical solidity and consistency; As well as the proposed theoretical model has been aligned with the theories and models that allow better writing, structure and directionality of the hypotheses. You can verify the information in the final version of the manuscript.

Instrument sources should be clearly stated as I think the items are not measuring the variables in the model.

  1. Answer: Thank you very much for your suggestion and observation. In the variable measurement section, the items that measure each variable have been described and the precursor authors who were used for their construction have also been added. You can verify the information in the final version of the manuscript.

Analysis has a problem of discriminant validity as the HTMT ratios are very high with one over 0.90 which points to serious problem of discriminant validity.

  1. Answer: Thank you very much for your suggestion and observation. Please inform us that in our discriminant validity results using the HTMT test, we did not find values above 9. The values in the first column correspond to the value of the average variance extracted (AVE), but they are not part of the values of the constructs to determine discriminant validity. You can verify the information in the final version of the manuscript.

Discussions need to compare and contrast with the literature.

  1. Answer: Thank you very much for your suggestion and observation. In the results section specifically in the discussions, we have presented the arguments for our findings, and they have been compared with both theory and previous studies. You can review this in the new version of the manuscript.

Implications need to be implementable.

  1. Answer: Thank you very much for your suggestion and observation. In the results section, specifically in the implications, we have listed a series of recommendations that can be implemented by those responsible for managing entrepreneurship in universities; however, these implications are not one hundred percent controllable by the researcher. You can review this in the new version of the manuscript.

Regards

Reviewer 3 Report

Comments and Suggestions for Authors

The article appears to be a promising contribution to the field of entrepreneurship studies, offering valuable insights into the influence of personal traits and psychological profiles on entrepreneurial intentions among university women.

Overall, whereas the introduction covers a wide array of factors relevant to female university entrepreneurship, it could significantly benefit from a clearer structure, focused direction, and concise articulation of its objectives: 

The introduction is lengthy and somewhat convoluted, making it challenging for readers to grasp the core message immediately. It delves into various aspects without a clear directional flow, leading to potential confusion.

While the introduction references several studies, it could improve by citing more recent and diverse sources, especially in a rapidly evolving field like entrepreneurship.

The introduction lacks a precise focus on the research problem or the main objective. It touches on numerous themes like the evolving role of women, their challenges, personality traits, psychological factors, and the impact of COVID-19, making it challenging to identify the central theme.

While discussing the impact of personality traits on entrepreneurship, the connection to the specific relevance for female university entrepreneurs and how it differs from the broader entrepreneurial context could be better articulated.

The introduction lacks smooth transitions between ideas, resulting in disjointed sections. It would benefit from a clearer organization that introduces each topic and its relevance to the study more cohesively.

While the introduction covers a wide array of factors and theories, it may need to emphasize the most critical aspects directly related to the study's purpose and research questions.

To draw the reader in, consider starting with a captivating hook that introduces the relevance of the research topic, possibly by stating a surprising statistic, real-world problem, or recent event related to female university entrepreneurship.

Strive for a balance between providing necessary background information and maintaining brevity. Focus on including the most pertinent information directly related to the study's scope and objectives.

The literature review and hypothesis development section provide a comprehensive analysis of various factors influencing the entrepreneurial intentions of university women. However, there are several aspects that could benefit from critical evaluation and improvement: refining its focus, ensuring clarity, reducing repetition, and explicitly linking findings to hypotheses could significantly strengthen the section's overall impact and cohesiveness. Here some suggestions: 

As the literature review references several studies to support the hypotheses, it could benefit from including more recent and diverse sources to strengthen the credibility and relevance of the claims made. Incorporating a broader range of perspectives and recent studies would enrich the review.

The section lacks a clear structure and cohesive flow, making it challenging for readers to follow the logical progression of ideas. The literature review should have a clearer delineation between sections and a more structured approach to present arguments and evidence.

There is some redundancy in the explanations and citations, with multiple studies reiterating similar points about personality traits, psychological profiles, and their influence on entrepreneurial intentions. Streamlining the content to avoid repetition could enhance the readability and conciseness of the review.

While the review discusses various factors related to personality traits, psychological profiles, and entrepreneurial intentions of university women, the direct link to each hypothesis could be more explicitly stated. Each hypothesis should be more tightly connected to the evidence presented in the literature review.

The literature review encompasses a broad spectrum of factors influencing entrepreneurship among women in university settings. Focusing on specific aspects directly related to the hypotheses could strengthen the review's relevance and coherence.

There's an opportunity to synthesize the findings of the reviewed literature more effectively. A more integrated discussion that draws connections between different studies, theories, and their collective implications on the hypotheses would strengthen the overall argument.

The transition from the literature review to the formulation of hypotheses could be made smoother. Explicitly connecting the key findings of the literature review to each hypothesis would reinforce the rationale behind each hypothesis.

The Results section is well-structured, providing a detailed analysis of the measurement and structural models. To improve, the section could benefit from a deeper interpretation of the statistical findings and a more integrated discussion between the measurement and structural model results.

Discussion and conclusion: Both sections effectively tie the study's results to existing theories and empirical evidence, providing valuable insights into the influence of personal traits and psychological profiles on entrepreneurial intentions among university women. However, enhancing clarity, language accessibility, and depth in addressing limitations could further strengthen the overall discussion and conclusion. For example:

- The discussion section, presented in a language other than English, might hinder comprehension for a wider audience, impacting the study's accessibility and reach (lines 484, 485, 486);

- As limitations are acknowledged, they could benefit from a more detailed exploration of their potential impact on the study's findings and broader implications.

- Although the conclusion highlights future research areas, it could provide more specific guidance or hypotheses for the direction of subsequent studies.

Author Response

(The authors gave the same response as above.)

Round 2

Reviewer 1 Report

Comments and Suggestions for Authors

Dear authors,

I am disappointed you didn't take my recommendations seriously after I put in several hours reviewing your paper. Most of the mistakes I mentioned earlier are still present. I explicitly told you to take care of punctuation issues etc. Even in the list of authors you forgot a comma after the second author and put a comma after the last author. This is just one example of your sloppiness. It seems to me you do not care. I don’t want to be associated with a low-quality paper being published under my watch. I gave you the benefit of the doubt and hoped you would improve the paper sufficiently. Yet, you failed horribly. I have to recommend a reject decision. 

All the best!

Comments on the Quality of English Language

Very poor English and even some mistakes I pointed at remained in the revised document.

Author Response

Dear reviewer, the authors send you an affectionate greeting and wish you the best for this year 2024. We thank you again for your extraordinary comments.

I am disappointed you didn't take my recommendations seriously after I put in several hours reviewing your paper. Most of the mistakes I mentioned earlier are still present. I explicitly told you to take care of punctuation issues etc. Even in the list of authors you forgot a comma after the second author and put a comma after the last author. This is just one example of your sloppiness. It seems to me you do not care. I don’t want to be associated with a low-quality paper being published under my watch. I gave you the benefit of the doubt and hoped you would improve the paper sufficiently. Yet, you failed horribly. I have to recommend a reject decision. 

Answer:

We inform you that we have made the recommended corrections. We have made these editorial and spelling corrections through a style correction expert from the MPDI publishing house.

Changes throughout the document regarding the content of all sections can be reviewed in the final version of the manuscript. These changes are marked in yellow.

Again, we appreciate your time and valuable contribution.

Regards

Reviewer 2 Report

Comments and Suggestions for Authors

The revisions are acceptable and the paper flows much better now.

Comments on the Quality of English Language

Acceptable.

Author Response

Dear reviewer, the authors send you a warm greeting and wish you the best for this year

2024. We appreciate your comments and observations.

We inform you that we have made the recommended corrections. We have made these

editorial and spelling corrections through a language style correction expert from the

MPDI publishing house.

Changes throughout the document regarding the content of all sections can be reviewed

in the final version of the manuscript. These changes are marked in yellow.

Again, we appreciate your time and valuable contribution.

Regards

Reviewer 3 Report

Comments and Suggestions for Authors

The changes made by the authors have enhanced the quality of the article, aligning with the suggestions and recommendations.

Author Response

(The authors gave the same response as above.)
